# Selective Formation of Unsymmetric Multidentate Azine-Based Ligands in Nickel(II) Complexes

**DOI:** 10.3390/molecules27206788

**Published:** 2022-10-11

**Authors:** Kennedy Mawunya Hayibor, Yukinari Sunatsuki, Takayoshi Suzuki

**Affiliations:** 1Graduate School of Natural Science and Technology, Okayama University, Okayama 700-8530, Japan; 2Advanced Science Research Center, Okayama University, Okayama 700-8530, Japan; 3Research Institute for Interdisciplinary Science, Okayama University, Okayama 700-8530, Japan

**Keywords:** (pyridyl)(imidazolyl)azines, aldazines, kryptoracemate, crystal structure

## Abstract

A mixture of 2-pyridine carboxaldehyde, 4-formylimidazole (or 2-methyl-4-formylimidazole), and NiCl_2_·6H_2_O in a molar ratio of 2:2:1 was reacted with two equivalents of hydrazine monohydrate in methanol, followed by the addition of aqueous NH_4_PF_6_ solution, afforded a Ni^II^ complex with two unsymmetric azine-based ligands, [Ni(HL^H^)_2_](PF_6_)_2_ (**1**) or [Ni(HL^Me^)_2_](PF_6_)_2_ (**2**), in a high yield, where HL^H^ denotes 2-pyridylmethylidenehydrazono-(4-imidazolyl)methane and HL^Me^ is its 2-methyl-4-imidazolyl derivative. The spectroscopic measurements and elemental analysis confirmed the phase purity of the bulk products, and the single-crystal X-ray analysis revealed the molecular and crystal structures of the Ni^II^ complexes bearing an unsymmetric HL^H^ or HL^Me^ azines in a tridentate κ^3^
*N*, *N’*, *N”* coordination mode. The HL^H^ complex with a methanol solvent, **1**·MeOH, crystallizes in the orthorhombic non-centrosymmetric space group *P*2_1_2_1_2_1_ with Z = 4, affording conglomerate crystals, while the HL^Me^ complex, **2**·H_2_O·Et_2_O, crystallizes in the monoclinic and centrosymmetric space group *P*2_1_/*n* with Z = 4. In the crystal of **2**·H_2_O·Et_2_O, there is intermolecular hydrogen-bonding interaction between the imidazole N–H and the neighboring uncoordinated azine-*N* atom, forming a one-dimensional polymeric structure, but there is no obvious magnetic interaction among the intra- and interchain paramagnetic Ni^II^ ions.

## 1. Introduction

Azines are a class of organic molecules with diimine functionality, R^1^R^2^C=N–N=CR^3^R^4^, which are often regarded as analogs of 1,3-butadiene (R^1^R^2^C=CH–CH=CR^3^R^4^) due to the resemblance and similarity in their functional groups [1,2], and represent a well-known class of compounds with interesting chemical properties [3] and applications in several and diverse fields [2]. In the area of synthetic organic chemistry, they serve as excellent synthons for obtaining heterocyclic compounds such as purines, pyrazoles, and pyrimidines by undergoing [2,3] criss-cross cycloaddition in the presence of dienophiles [4,5,6]. They are good pharmaceutical and biological agents, for example, antimalarial, anticonvulsant, and antioxidant properties [7,8,9,10]. In the agricultural field, azines exhibit herbicidal properties [11,12]. Azine-based polymers are successfully used for chemosensors [13,14,15] and they also possess promising cathodic abilities for organic batteries [16,17,18] and could be also employed for 2D field effect transistors in their crystallized phase as silicon [19]. Some azine-based polymers possess promising cathodic abilities for organic batteries and are vital in optoelectronic devices. Furthermore, compounds containing the diimine (C=N–N=C) linkers have been investigated as highly luminescent frameworks [20,21,22] as well as good photosensitizers in solar cells [23].

The coordinating ability of the diimine group in the azines is interesting due to the flexibility of the N–N bond. A series of transition-metal complexes of azine-based ligands, which possess interesting structural motifs and other properties, have been reported [2,24]. Pyridyl ketazine is one of the unique azine-based ligands that have been explored by several researchers for its coordinating abilities and modes, thus paving way for the investigation of several azine-based ligands for their unique coordination chemistry [25,26,27]. Our previous studies reported a series of mono- and dinuclear iron(II) complexes with imidazole-4-carbaldehyde azine and other imidazole groups as azine ligands showcasing different modes of coordination towards the iron(II) in the various complexes [28,29,30].

Recently, we have investigated the chemistry of pyridyl and imidazolyl azine-based ligands, where an unprecedented selective synthesis of the unsymmetric (2-pyridyl)(2-methyl-4-imidazolyl)azine is obtained in an excellent yield with an interesting bonding mode [30]. Pursuing our interest in the unsymmetric azine-based ligand, we attempted to synthesize the corresponding nickel(II) complexes bearing 2-pyridylmethylidenehydrazono-(4-imidazolyl)methane (HL^H^) and 2-pyridylmethylidenehydrazono-(2-methyl-4-imidazolyl)methane (HL^Me^).

## 2. Results and Discussion

### 2.1. Preparation of Unsymmetric (2-Pyridyl)(4-Imidazolyl)azines and Their Nickel(II) Complexes

The most intuitive and possibly simplest method for preparation of an unsymmetric (2-pyridyl)(4-imidazolyl)azine is a reaction of stoichiometric amounts of 2-pyridinecarboxaldehyde, 1*H*-imidazole-4-carboxaldehyde (or its 2-methyl derivative), and hydrazine (Figure 1) [31,32]. However, as was expected, all attempts to prepare the compounds HL^R^ with this method failed to isolate the desired compounds, because the reaction gave a complicated mixture of the products (Appendix A) which were hard to be separated by any purification method. In a previous study [30], we serendipitously found that the reaction in the presence of iron(II) salts gave selectively the crystals of a Fe^II^ complex bearing unsymmetric azine, [Fe(HL^Me^)_2_](PF_6_)_2_·1.5H_2_O. To clarify the role of transition-metal salts in the selective formation of a certain complex, we used a nickel(II) chloride for the preparation of (2-pyridyl)(4-imidazolyl)azine complexes.

A mixture of 2-pyridine carboxaldehyde, 4-formylimidazole (or 2-methyl-4-formylimidazole), and NiCl_2_^.^6H_2_O in a 2:2:1 molar ratio in methanol was reacted with a stoichiometric amount of hydrazine monohydrate, followed by the addition of an aqueous solution of NH_4_PF_6_, which gave an obvious color change of the reaction solution to deep reddish orange. From the reaction mixture, air-stable deep reddish orange crude product (**1** from 4-formylimidazole or **2** from the 2-methyl derivative) was obtained by evaporation of the solvent in a relatively high yield (80% and 83% for compounds **1** and **2**, respectively). The crude products are soluble in common polar organic solvents and recrystallized from acetonitrile by vapor diffusion of methanol to deposit block-shaped deep reddish orange crystals of **1**·MeOH. For compound **2**, platelet single-crystals of **2**·H_2_O·Et_2_O suitable for X-ray diffraction study were deposited by vapor diffusion of diethyl ether into a methanol solution. In the FT-IR measurement of both compounds, the crude and recrystallized products gave almost identical spectra, which showed ν(C=N) stretching bands at 1619 and 1603 cm^−1^ for **1** and 1625 and 1603 cm^−1^ for **2** (Appendix A). This suggests that like the above-mentioned Fe^II^ complex [30], a certain Ni^II^ complex was selectively formed among several possible products. The elemental analyses of the vacuum-dried (partially efflorescent) samples suggested the empirical composition of [Ni(HL^H^)_2_](PF_6_)_2_·0.5MeOH and [Ni(HL^Me^)_2_](PF_6_)_2_^.^MeCN^.^1.5MeOH for **1** and **2**, respectively.

The χ_M_*T* values of **1** and **2** at 300 K are 1.14 and 1.21 cm^3^ K mol^−1^, respectively. These values are almost constant down to 20 K, then, decrease sharply below 20 K due to magnetic anisotropies of them (Appendix A). No significant magnetic interactions between complex cations were observed. In addition, magnetizations at 1.9 K for both complexes (Appendix A) did not reach the saturation values at 5 T, indicating the existence of magnetic anisotropies for both complexes. They are common behavior for magnetically isolated octahedral mononuclear nickel(II) complexes.

Absorption spectra of complexes **1** and **2** recorded in acetonitrile at room temperature were shown in Appendix A. Both complexes displayed two absorption bands in the region of 200–550 nm. The absorption bands in the higher energy region around 200–330 nm can be assigned to ligand-centered (LC) π–π* and n–π* transitions, respectively. The lowest energy absorption band for the complexes around 450–550 nm can be ascribed as the metal-to-ligand charge transfer (MLCT) band.

### 2.2. Crystal Structures of the Nickel(II) Complexes

The molecular and crystal structures of **1**·MeOH and **2**·H_2_O·Et_2_O was confirmed by the single-crystal X-ray analysis at 188(2) K. Compound **1**·MeOH crystallized in the orthorhombic system and a non-centrosymmetric space group *P*2_1_2_1_2_1_ with Z = 4 (Table 1), indicating conglomerate crystallization (spontaneous resolution of the enantiomers). The asymmetric unit consists of one [Ni(HL^H^)_2_]^2+^ cation, two PF_6_^−^ anions, and a methanol molecule of crystallization. An ORTEP drawing of **1** is shown in Figure 1. The Ni^II^ center was coordinated by two HL^H^ ligands in a pseudo-octahedral coordination geometry. Each HL^H^ ligand has an *E*,*Z* configuration (mode (i) in Figure 2) serving as tridentate coordination to a Ni^II^ center in a meridional fashion via pyridyl-*N*, imidazolyl-*N*, and one of the azine-*N* atoms close to the pyridyl substitution group. This coordination mode forms a five-membered chelate ring on the pyridine side and a six-membered one on the imidazole side. It is noted that the other azine-*N* atom remains uncoordinated and the imidazole-NH group remains protonated.

The coordination bond lengths and angles of **1** are summarized in Table 2, which indicates the nearly ideal octahedral coordination geometry around the Ni center, with minor deviations. The Ni–N bond lengths are in the range of 2.039(5)–2.095(5) Å, which are typical for Ni^II^–N(imine) coordination bonds [33,34]. The five-membered chelate bite angles (N1–Ni1–N2 and N6–Ni1–N7) are smaller by ca. 10° than the six-membered chelate bite angles (N2–Ni1–N5, N7–Ni1–N10), as expected. The mutually *trans* bond angle of N2–Ni1–N7 for the azine-*N* donors (175.7(2)°) is close to the ideal value.

The packing structure of **1**·MeOH was illustrated in Appendix A. In the crystal structure, an explicit hydrogen-bond was observed between one of the imidazole N–H group and the O atom of the methanol molecule of crystallization: N9(–H)···O1 2.710(8) Å (Figure 1), but no other intermolecular interactions were found. In a previous study, we reported the crystal structure of the analogous Fe^II^ complex, [Fe(HL^Me^)_2_](PF_6_)_2_·1.5H_2_O [30], in which a noble kryptoracemate resulted from a formation of a one-dimensional helical polymer by an intermolecular hydrogen-bonding interaction. In the present Ni^II^ complex **1**·MeOH, although the compound was crystallized in a non-enantiogenic (Sohncke) space group, *P*2_1_2_1_2_1_, the complex cation was crystallized in a discrete form (Appendix A) and did not show the kryptoracemate phenomenon. We have tried to measure the solid-state CD spectra of a piece of single-crystal of **1**·MeOH (in a KBr disk), but no CD signal was observed.

The compound, **2**·H_2_O·Et_2_O, crystallized in the monoclinic system and centrosymmetric space group *P*2_1_/*n* with Z = 4 (Table 1). The molecular structure of the Ni^II^ complex cation in **2** (Figure 2) is very similar to that in **1**, except for the large deviation of the bond angles, e.g., N2–Ni1–N7 and N5–Ni1–N7 (Table 2), which resulted from steric congestion from the substituted methyl group at the imidazole ring.

In the crystal structure, the intermolecular hydrogen-bonding interaction was observed between the imidazole N–H and azine-N groups: N9(–H9)···N8 2.714(6) Å, forming one-dimensional coordination polymers (Figure 3). In contrast to the corresponding Fe^II^ complex [30], this Ni^II^ complex **2**·H_2_O·Et_2_O crystallized in a centrosymmetric space group *P*2_1_/*n*, indicating the crystal consists of the racemic mixture.

## 3. Materials and Methods

### 3.1. Chemicals and Physical Methods

All chemicals and solvents used for syntheses of azine compounds and Ni complexes were reagent grade and used without further purification. First, 2-pyridinecarboxaldehyde, 1*H*-imidazole-4-carboxaldehyde, 2-methyl-1*H*-imidazole-4-carboxaldehyde, nickel(II) chloride hexahydrate, and ammonium hexafluorophosphate were purchased from FUJIFILM (Tokyo, Japan). All reactions were carried out under aerobic conditions. Infrared spectra (KBr pellets; 4000–400 cm^−1^) were recorded on a JASCO FT-001 Fourier transform infrared spectrometer (JASCO, Tokyo, Japan). Absorption spectra were recorded on a Shimadzu UV/Vis-1650 spectrophotometer (Kyoto, Japan) in the range of 200–600 nm at room temperature in acetonitrile. The ^1^H NMR spectra were acquired on a Varian 400-MR spectrometer (Los Angeles, CA, USA); the chemical shifts were referenced to residual ^1^H NMR signals of solvents and are reported versus TMS. Elemental analyses were conducted at Advanced Science Research Center, Okayama University. Magnetic susceptibilities were measured on a Quantum Design MPMS XL5 SQUID magnetometer (Tokyo, Japan) in a 1.9–300 K temperature range under an applied magnetic field of 0.1 T at the Okayama University of Science. Corrections for diamagnetism were applied using Pascal’s constants [35].

### 3.2. Preparation of Nickel(II) Complexes

#### 3.2.1. [Ni(HL^H^)_2_](PF_6_)_2_ (**1**)

A methanol solution (30 mL) of NiCl_2_·6H_2_O (0.477 g, 2.00 mmol) was added to a methanol solution (60 mL) containing 2-pyridinecarboxaldehyde (0.432 g, 4.00 mmol) and 1*H*-imidazole-4-carboxaldehyde (0.387 g, 4.00 mmol), followed by additions of hydrazine monohydrate (0.207 g, 4.00 mmol) in methanol (30 mL) and NH_4_PF_6_ (0.652 g, 4.00 mmol) in water (20 mL). The mixture was stirred at ca. 60 °C for 3 h. The resulting solution was concentrated by a rotary vacuum evaporator to give a deep reddish-orange precipitate. The crude product was dissolved in methanol and acetonitrile and recrystallized by slow evaporation to deposit deep reddish-orange microcrystals. Crystals suitable for the SC-XRD study were obtained from a mixture of acetonitrile and methanol. Yield: 1.72 g (80%). Anal. Found: C, 32.41; H, 2.36; N, 18.38%. Calcd for C_20.5_H_20_F_12_NiN_10_O_0.5_P_2_ (for **1**·0.5MeOH: C, 32.27; H, 2.64; N, 18.36%. IR (KBr pellet): ν_C=N_ (imine) 1619, 1603 cm^−1^; ν_P–F_ (PF_6_^−^) 840 cm^−1^.

#### 3.2.2. [Ni(HL^Me^)_2_](PF_6_)_2_ (**2**)

Complex **2** was obtained in a similar manner using 2-methyl-1*H*-imidazole-4-carboxaldehyde instead of 1*H*-imidazole-4-carboxaldehyde. Yield: 83%. Anal. Found: C, 35.45; H, 3.43; N, 17.95%. Calcd for C_25.5_H_31_F_12_NiO_1.5_P_2_ (for **2**·CH_3_CN·1.5CH_3_OH: C, 35.44; H, 3.62; N, 17.83. IR (KBr pellet cm^−1^): ν_C=N_ (imine) 1635, 1609 (fs) ν_P–F_ (PF_6_^−^) 845(s). Deep reddish-orange platelet crystals (**2**·H_2_O·Et_2_O) suitable for SC-XRD were obtained from a mixture of methanol and diethyl ether.

### 3.3. Structure Determination by X-ray Crystallography

The single-crystal X-ray diffraction data for compounds **1**·MeOH and **2**·H_2_O·Et_2_O were collected at 188(2) K using a Rigaku RAXIS RAPID II imaging plate area detector employing graphite monochromated Mo K*α* radiation (λ = 0.71073 Å). The structures were solved by the direct method, employing the SIR2014 software packages [36], and refined on *F*^2^ by full-matrix least-squares techniques using the SHELXL2014 program package [37]. All non-hydrogen atoms were refined anisotropically, and hydrogen atoms were included in the calculations with riding models. All calculations were performed using the Crystal Structure software package [38]. The crystal parameters, data collection procedure, and refinement results for the two compounds **1**·MeOH and **2**·H_2_O·Et_2_O are summarized in Table 1.

## 4. Conclusions

In this study, we attempted to prepare transition-metal(II) complexes of an unsymmetrical azine-type ligand, HL^R^, having 2-pyridyl and (2-methyl-)1*H*-imidazol-4-yl substituent groups. The desired azine could not be isolated in pure form from a simple stoichiometric reaction of hydrazine and respective aldehydes. However, in our previous study using Fe^II^ salts, a highly selective formation of [Fe(HL^Me^)_2_](PF_6_)_2_·1.5H_2_O was observed, and the complex was found to be a kryptoracemate as a result of a one-dimensional helical chain structure by hydrogen-bonding interaction. At present, we have studied another two cases with nickel(II) salts: [Ni(HL^H^)_2_](PF_6_)_2_·MeOH (**1**·MeOH) and [Ni(HL^Me^)_2_](PF_6_)_2_·H_2_O·Et_2_O (**2**·H_2_O·Et_2_O). In both cases, a highly selective formation of the unsymmetrical azine complex was observed among other possible symmetrical and/or unsymmetrical complexes.

In the crystal of **1**·MeOH, the compound was crystallized in a non-enantiogenic (Sohncke) space group, *P*2_1_2_1_2_1_, but the complex cation, [Ni(HL^H^)_2_]^2+^, was only hydrogen-bonded to the solvated methanol molecule. In the crystal structure of **2**·H_2_O·Et_2_O, there observed a one-dimensional hydrogen-bonded polymer chain made from [Ni(HL^Me^)_2_]^2+^, but it was crystallized in a centrosymmetric space group, *P*2_1_/*c*. Thus, it can be concluded that the reason for the selective formation of an unsymmetric azine ligand in [Fe(HL^Me^)_2_](PF_6_)_2_·1.5H_2_O was not solely the formation of the characteristic hydrogen-bonded chain. The suitable tridentate chelate formation of *E,Z*-HL^R^ with mode (i) (in Figure 2), which gives a five-membered chelate ring at the pyridyl coordination site and a six-membered chelate ring at the imidazolyl one, would probably be the most stable among the other coordination modes of symmetrical and unsymmetrical azine derivatives.

## Data Availability

Crystallographic data for compounds **1**·MeOH and **2**·H_2_O·Et_2_O have been deposited with the Cambridge Crystallographic Data Centre, CCDC 2209932, and 2209933, respectively. These data can be obtained free of charge from The Cambridge Crystallographic Data Centre via www.ccdc.cam.ac.uk/data_request/cif (accessed on 28 September 2022).

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
