# Peer review of "Selective Formation of Unsymmetric Multidentate Azine-Based Ligands in Nickel(II) Complexes"

_molecules, 2022, doi:10.3390/molecules27206788_

Round 1
Reviewer 1 Report
The Authors presented a very interesting article with great opportunity of future investigation of azide based for optoelectronic devise as they mentioned.
The article is well written and I suggest to be accepted with minor reviews and here my comments.
1) At line 35 you mentioned that azine-based polymenrs can be used for batteries and for optoelectronic devices. I suggest to reformulate a little bit this phrase in this way: “Azine-based polymers are successfully used for chemosensors [put some references, please]and they also possess promising cathodic abilities for organic batteries [put some references, please] and could be also employed for 2D field effect transistors in their crystallized phase as silicon [cite: https://doi.org/10.1021/acsnano.0c10609].”
2) The Figure 2 must be substituted with one in higher definition and avoiding the head hake of reader. You have wonderfully showed the ellipsoid that show the atomic vibration as function of temperature. Actually, the show the high quality of your results. Here in this Figure 2 you must be mush clear as you can. Please!
Author Response
Reviewer #1: The Authors presented a very interesting article with great opportunity of future investigation of azide based for optoelectronic devise as they mentioned.
The article is well written and I suggest to be accepted with minor reviews and here my comments.
We thank this reviewer for his/her positive recommendation. According to the comments given by this reviewer, we have revised our manuscript as attached.
—————————————————————————————————————
1) At line35 you mentioned that azine-based polymenrs can be used for batteries and for optoelectronic devices. I suggest to reformulate a little bit this phrase in this way: “Azine-based polymers are successfully used for chemosensors [put some references, please] and they also possess promising cathodic abilities for organic batteries [put some references, please] and could be also employed for 2D field effect transistors in their crystallized phase as silicon [cite: https://doi.org/10.1021/acsnano.0c10609].”
Thank you for the suggestion for revision. We agreed to the comment, and revised the text accordingly. In addition, we have also added a sentence: In the agricultural field, azines exhibit herbicidal properties, just before the above-revision.We have added some references, and therefore, changed the numbers of the references afterwards.
—————————————————————————————————————
2) The Figure 2 must be substituted with one in higher definition and avoiding the head hake of reader. You have wonderfully showed the ellipsoid that show the atomic vibration as function of temperature. Actually, the show the high quality of your results. Here in this Figure 2 you must be mush clear as you can. Please!
Thank you for this suggestion. We have replaced this figure (packing diagram) to the supplementary material (as Figure S5). The numbers of the figures afterwards were changed appropriately.
Reviewer 2 Report
- - The role of metal ion in forming the ligand is really important, can you provide an explanation or rationale mechanism.
- - Account of level B error in the crystal of Nicomp1.
- - Account of level B errors in the crystal of Nicomp2
Author Response
Reviewer #2: - The role of metal ion in forming the ligand is really important, can you provide an explanation or rationale mechanism.
Yes, it is. We have added a little explanation more to the reason why the selective formation of the unsymmetric azine in this system was preferable in the last part of Conclusion.
—————————————————————————————————————
- Account of level B error in the crystal of Nicomp1.
The calculated Flack parameter is 0.022(7), which is not so higher than 0, as compared to the final Rvalues. We have checked the inverted structure, which indicated that the reported absolute conformation would be correct.
—————————————————————————————————————
- Account of level B errors in the crystal of Nicomp2
Because of slightly insufficient quality of the diffraction data, the thermal ellipsoids of some atoms (e.g., C13, C23, etc.) are relatively high, but we believe the analytical result was enough acceptable. Also, we did not include the H atoms of the hydrated water molecule (O1), which has also a large thermal ellipsoid, the calculated O•••O distance became relatively short.